# MoRL: Reinforced Reasoning for Unified Motion Understanding and Generation

## Abstract

Human motion understanding and generation are crucial for vision and robotics but remain limited in reasoning capability and test-time planning. We propose MoRL, a unified multimodal motion model trained with supervised fine-tuning and reinforcement learning with verifiable rewards. Our task-specific reward design combines semantic alignment and reasoning coherence for understanding with physical plausibility and text–motion consistency for generation, improving both logical reasoning and perceptual realism. To further enhance inference, we introduce Chain-of-Motion (CoM), a test-time reasoning method that enables step-by-step planning and reflection. We also construct two large-scale CoT datasets, MoUnd-CoT-140K and MoGen-CoT-140K, to align motion sequences with reasoning traces and action descriptions. Experiments on HumanML3D and KIT-ML show that MoRL achieves significant gains over state-of-the-art baselines.

## 1 Introduction

Human motion understanding and generation are fundamental problems in computer vision and robotics. They enable a wide range of applications, from interactive character animation and robotics to game development and virtual reality. With the advent of large-scale motion capture datasets and expressive parametric human models such as SMPL (Loper et al., 2023) and SMPL-X (Pavlakos et al., 2019), recent years have witnessed rapid progress in text-to-motion generation (Jiang et al., 2023; Guo et al., 2022a; Gong et al., 2023) and motion-language alignment (Zhang et al., 2023a; Guo et al., 2022a). Currently, the success of large language models (LLMs) has inspired multimodal extensions that integrate text, image, and 3D signals, pushing the frontier of motion language modeling toward more scalable and generalizable systems. Existing approaches have begun to explore this space. MotionGPT (Jiang et al., 2023) considers motion as a foreign language to establish a unified action language framework. MotionRL Liu et al. (2024) introduces multi-reward optimization to better match human preferences. More recently, Motion-R1 (Ouyang et al., 2025) applies Chain-of-Thought reasoning and reinforcement learning to motion generation.

Despite these advances, two major challenges remain. First, current models treat user queries as a whole, with limited reasoning capability. They struggle to parse prompts into fine-grained steps or to understand or generate detailed motions in a step-by-step manner. Second, at test time, most models simply decode outputs in a single pass. They lack explicit planning or reflection, and therefore cannot fully exploit the reasoning ability of large language models.

To address the first challenge, we propose **MoRL**, a multimodal motion unified model that unifies motion understanding and generation under a reinforcement learning framework. MoRL is trained with a hierarchical post-training pipeline. We then perform reinforcement learning with verifiable rewards (RLVR). Unlike prior works that rely primarily on generic similarity scores, our reward design is task-specific and dual-headed: for motion understanding, we introduce semantic alignment and a novel reasoning coherence reward that enforces logically consistent reasoning traces; for motion generation, we combine text–motion consistency with a physical plausibility reward that enforces biomechanical validity. This combination provides a simple yet innovative way to align model outputs with both semantic fidelity and human perceptual realism.

To address the second challenge, and improve the test-time performance, we introduce Chain-of-Motion (CoM), a decoding strategy that explicitly incorporates step-by-step reasoning and reflection. CoM not only improves the robustness of reasoning-based motion understanding but also refines motion generation through iterative selection and correction. Moreover, the same principle guides the synthesis of our CoT datasets, ensuring consistency between training and inference. Specifically, we construct two large-scale synthetic Chain-of-Thinking (CoT) datasets, MoUnd-CoT-140K and MoGen-CoT-140K, to align motion sequences with reasoning traces and concise action descriptions.

To further showcase the effectiveness, we conduct comprehensive experiments on HumanML3D (Guo et al., 2022a) and KIT-ML (Plappert et al., 2016). Results show that MoRL achieves significant gains over SOTA baselines.

In summary, the main contributions are:

- We propose **MoRL**, a unified multimodal motion model that combines semantic alignment and reasoning-coherence rewards for motion understanding with physical plausibility and text–motion consistency rewards for motion generation, effectively improving logical reasoning and alignment with human perceptual realism.

- We introduce Chain of Motion, a test-time reasoning method, along with two large-scale CoT datasets, MoUnd-CoT-140K and MoGen-CoT-140K, which enhance motion understanding and generation through step-by-step reasoning and reflection.

- Extensive experiments on HumanML3D and KIT-ML show that MoRL substantially outperforms state-of-the-art baselines, achieving a 4.17% improvement in BERT scores and 3% improvement in FID, respectively.

## 2 RELATED WORKS

**Motion understanding and generation.**   Recent work on human motion understanding and generation has rapidly evolved from specialized sequence models to large language model (LLM)–based frameworks that unify perception, reasoning, and text–motion alignment. Early multimodal approaches such as MotionLLM (Chen et al., 2024), ChatPose (Feng et al., 2024), and ChatHuman (Lin et al., 2024) explored conversational or interactive motion generation, yet their evaluations largely focused on qualitative results without systematic motion-to-text benchmarking. UniMotion (Li et al., 2025) extended cross-modal modeling to a broader set of human activities, but it similarly omitted explicit motion-to-text evaluation, leaving the bidirectional mapping under-explored. LLM-driven pipelines such as MotionLLaMA (Ling et al., 2024) demonstrated impressive compositional motion synthesis but relied on private datasets, limiting reproducibility and large-scale comparison. Structured agent architectures like ACMo and CoMA (Sun et al., 2024) further highlighted the benefits of compositional reasoning and multi-modal interaction for controllable human-motion generation. Building on these foundations, a new wave of motion-generation systems integrates transformer backbones with LLM reasoning. Representative examples include MotionGPT (Zhang et al., 2024d; Ribeiro-Gomes et al., 2024), T2M-GPT (Wang, 2023), and ReMoGPT (Yu et al., 2025), which leverage powerful language priors to improve both motion synthesis and natural-language controllability. Despite these advances, unified evaluation protocols that cover motion-to-text understanding, text-conditioned generation, and open-dataset benchmarking remain limited, motivating the need for methods that jointly address generation fidelity and cross-modal reasoning.

**Large language model reasoning.**   Many studies aim to enhance the reasoning capacity of Large Language Models (LLMs) to perform complex, multi-step problem-solving tasks by employing Chain-of-Thought (CoT) prompting (Wei et al., 2022; Zhang et al., 2023b; 2024c; Mitra et al., 2024; Hao et al., 2024; Yao et al., 2023; Yuan et al., 2024a; Luan et al., 2024) and conducting supervised fine-tuning (SFT) with step-level supervision (Zhang et al., 2024a; Zhao et al., 2024; Yao et al., 2024; Thawakar et al., 2025). Recently, DeepSeek-R1 (Guo et al., 2025) successfully applied rule-based Reinforcement Learning (RL) (Shao et al., 2024) to induce the self-emergence of complex cognitive reasoning abilities in LLMs, demonstrating that even coarse, outcome-only rewards can effectively elicit strong reasoning behavior. Its success demonstrated that, with a carefully designed reward structure and policy optimization strategy, models can learn to generate long CoT reasoning without the need for intermediate supervision. Building on this paradigm, recent efforts such

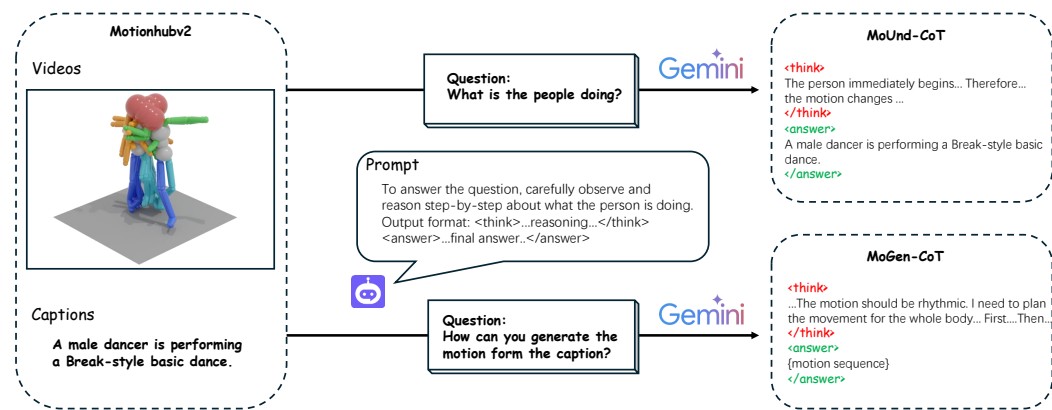

Figure 1: **Motion CoT Data Engine.** Build based on MotionHubV2 dataset (Ling et al., 2024), one branch (MoUnd-CoT) uses motion sequences and captions with Gemini to construct reasoning chains for understanding, while the other (MoGen-CoT) builds reasoning chains for generation.

as Open-Reasoner-Zero (Hu et al., 2025) and Kimi k1.5 (Team et al., 2025) have adopted similar rule-based reinforcement learning pipelines to enhance reasoning in the text and image domains, respectively. However, despite these promising developments, little prior work has investigated extending this approach to the video domain. Bridging this gap remains both a significant challenge and a promising direction for advancing the capabilities of reasoning models.

# 3 DATA SYNTHESIS

**Data engine.** The key to empowering MoRL with strong reasoning ability lies in large-scale, high-quality chain-of-thought (CoT) data. To address this gap, we design a data engine, as shown in Figure 1, built on Gemini-2.5-pro (Comanici et al., 2025). It performs gap-based reasoning through question–answer pairs and captures the reasoning process. This aligns motion sequences with natural language reasoning chains and concise action captions. The sequences and captions are derived from the MotionHubV2 dataset (Ling et al., 2024), which is constructed as a subset of multiple publicly available datasets and encompasses diverse motion scenarios such as dance, performance interaction and various activities from daily life. The resulting dataset consists of two complementary branches: Motion Understanding and Motion Generation. Together, they form a unified CoT resource.

**MoUnd-CoT-140K.** The motion understanding branch, denoted as *MoUnd-CoT-140K*, is designed to map motion sequences into textual reasoning and descriptive outputs. Each data sample contains three components: (i) a motion sequence represented in the standard SMPL-X format, (ii) a reasoning chain enclosed in `<think>` tags, and (iii) a concise caption of the action enclosed in `<answer>` tags. To ensure compatibility with HumanML3D-style features, we convert SMPL-X joint sequences into humanml joint sequences and then extract motion features of dimension 263 per frame. This allows the dataset to be directly consumed by existing motion-language models. The resulting MoUnd-CoT-140K dataset provides high-quality CoT supervision for motion understanding tasks, especially in scenarios where the model must first interpret motion dynamics before generating a compact description.

**MoGen-CoT-140K.** The motion generation branch, denoted as *MoGen-CoT-140K*, complements MoUnd-CoT-140K by focusing on the inverse process: generating motion sequences from textual reasoning and descriptive inputs. Each sample contains (i) a natural language caption of the intended action, (ii) an associated reasoning chain in `<think>` tags, and (iii) the corresponding motion sequence stored in SMPL-X format contained between `<answer>` tags. For consistency, all sequences are normalized into the HumanML3D feature space. MoGen-CoT-140K thus enables motion-language models to learn not only to understand motion but also to generate realistic, semantically aligned motion sequences guided by reasoning signals.

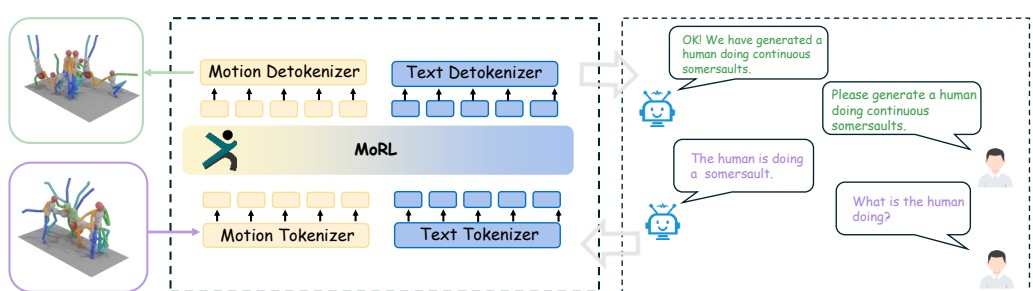

Figure 2: **Overview of MoRL.** Our framework unifies motion understanding and generation under a reinforcement learning paradigm. Motion and text inputs are tokenized into a shared representation space. A hierarchical post-training pipeline first applies supervised fine-tuning (SFT) on large-scale synthetic CoT datasets to align motion sequences with reasoning traces and concise descriptions, then employs reinforcement learning with verifiable rewards (RLVR) to refine outputs, enhancing semantic alignment, reasoning coherence, physical plausibility, and text–motion consistency. At inference, the Chain-of-Motion (CoM) decoding strategy enables step-by-step reasoning and reflection, improving both motion understanding and perceptually realistic motion generation.

Together, MoUnd-CoT-140K and MoGen-CoT-140K form a balanced CoT-based motion-language corpus, enabling instruction tuning for both understanding and generation within a unified framework.

## 4    THE PROPOSED METHOD

### 4.1    OVERVIEW

As shown in Figure 2, we propose MoRL, a multimodal motion foundation model that unifies both understanding and generation of human motion. The whole architecture is built upon a multimodal large language model (MLLM) initialized from Qwen3-4B-Instruct Yang et al. (2025), augmented with a text tokenizer and a motion tokenizer for modality alignment. The framework consists of three key components: (1) a supervised fine-tuning (SFT) stage with a synthetic CoT dataset for cold-start training; (2) reinforcement learning (RL) policies designed separately for motion understanding and motion generation, optimized with task-specific reward functions; and (3) a well-designed test-time method, Chain-of-Motion (CoM), which enhances both understanding and generation via structured step-by-step justification.

### 4.2    ARCHITECTURE

MoRL adopts a unified multimodal LLM backbone equipped with two modality-specific tokenizers. The text tokenizer is inherited from the base language model, while the motion tokenizer discretizes continuous 3D human motion into compact motion tokens via a VQ-VAE style encoder–decoder. The multimodal fusion is achieved through shared transformer layers, enabling cross-attention between textual and motion representations. This design follows the paradigm of motion-language alignment in Deepseek, but extends it to bidirectional tasks, including text-to-motion generation and motion-to-text understanding.

**Text tokenizer.**    We employ the native tokenizer of LLM to map natural language into subword tokens. This preserves the rich linguistic knowledge of the base LLM while ensuring compatibility with motion-related vocabulary introduced during supervised fine-tuning. The text tokens serve as both queries (in understanding tasks) and conditioning signals (in generation tasks).

**Motion tokenizer.**    To bridge the gap between continuous human motion and the discrete token space of the LLM, we adopt a VQ-VAE style motion tokenizer. Given an input motion sequence $m_{1:T} \in \mathbb{R}^{T \times D}$, where $T$ is the number of frames and $D$ is the dimensionality of each frame, the

encoder $E$ compresses the sequence into latent vectors $z_{1:(T/l)} \in \mathbb{R}^{(T/l) \times d}$ with downsampling factor $l$ and latent dimension $d$. Each latent $z_i$ is then quantized against a learnable codebook $\mathcal{C} = \{c_n\}_{n=1}^{N}$:

$$\hat{z}_i = \arg\min_{c_n \in \mathcal{C}} \|z_i - c_n\|_2^2. \tag{1}$$

The quantized sequence $\hat{z}_{1:(T/l)}$ is decoded back to reconstruct the original motion $\hat{m}_{1:T} = D(\hat{z}_{1:(T/l)})$. Training follows the composite VQ-VAE loss:

$$\mathcal{L}_{vq} = \mathcal{L}_{reconstruct} + \mathcal{L}_{commit} + \mathcal{L}_{embed}, \tag{2}$$

where $\mathcal{L}_{reconstruct}$ is a smoothed L1 loss with velocity regularization, $\mathcal{L}_{commit}$ enforces codebook utilization, and $\mathcal{L}_{embed}$ stabilizes latent representations. This discrete motion representation not only reduces sequence length but also aligns seamlessly with the autoregressive generation paradigm of LLMs.

## 4.3 COLD START STAGE

Recent work such as DeepSeek-R1 (Guo et al., 2025) demonstrated that reinforcement learning alone can sometimes induce CoT reasoning. Motivated by this, we first explored training our motion–language model directly with RL signals. In practice, however, this strategy was highly unstable: the model rarely produced well-formed reasoning traces and even generated answers that deviated from the intended semantics. To stabilize training, we introduce a cold-start phase based on supervised fine-tuning. Specifically, we use our synthetic datasets MoUnd-CoT-140K and MoGen-CoT-140K, which couple motion sequences with reasoning steps (`<think>`) and concise descriptions (`<answer>`). Supervised finetuning on these data forces the model to follow the required output format, stabilizing its outputs and ensuring semantic consistency between inference and final answers. This initialization greatly reduces collapse during RL and establishes a reliable starting point for policy optimization.

## 4.4 REINFORCEMENT LEARNING

After cold-start training, we further align the model outputs with task objectives through reinforcement learning. We adopt a group-based policy optimization strategy similar to GRPO, where multiple candidate outputs are sampled per prompt, scored with reward functions, normalized within the group, and used to compute policy gradients with a KL regularization term to a frozen reference model.

**Motion understanding.** For motion understanding, the model must output a reasoning trace $\hat{r}$ and a caption $\hat{a}$ given a motion sequence $m$. We define two rewards:

*Semantic Alignment Reward.* We measure the semantic similarity between $\hat{a}$ and the reference caption $a$ using a pretrained text encoder $E_{\text{text}}$:

$$R_{\text{sem}} = \cos(E_{\text{text}}(\hat{a}), E_{\text{text}}(a)). \tag{3}$$

*Reasoning Coherence Reward.* We encourage the reasoning trace to logically support the answer using an NLI model $f_{\text{NLI}}$:

$$R_{\text{coh}} = f_{\text{NLI}}(\hat{r}, \hat{a}), \tag{4}$$

where $f_{\text{NLI}}(\cdot)$ outputs an entailment confidence score.

**Motion generation.** For motion generation, the model produces a motion sequence $\hat{m}$ from a text prompt $t$. We use two rewards:

*Physical Plausibility Reward.* We penalize implausible motion dynamics:

$$R_{\text{phys}} = -\lambda_1 \cdot L_{\text{joint}}(\hat{m}) - \lambda_2 \cdot L_{\text{vel}}(\hat{m}), \tag{5}$$

where $L_{\text{joint}}(\cdot)$ measures joint-angle violations and $L_{\text{vel}}(\cdot)$ penalizes abrupt velocity changes.

*Text–Motion Consistency Reward.* We enforce semantic alignment between generated motion and the input text, using cross-modal encoders $E_{\text{text}}, E_{\text{motion}}$:

$$R_{\text{align}} = \cos(E_{\text{text}}(t), E_{\text{motion}}(\hat{m})). \tag{6}$$

**Discussion.** The final reward is a weighted sum of these components. Despite their simplicity, the four rewards cover both semantic and physical aspects of motion–language alignment. This MoRL stage thus provides an effective yet lightweight way to refine the model, avoiding overly complex heuristics while yielding consistent gains in both understanding and generation.

### 4.5 CHAIN-OF-MOTION (COM)

Most existing motion–language models decode outputs in a single forward pass, which often leads to semantically shallow reasoning in understanding tasks and temporally incoherent sequences in generation tasks. To overcome this limitation, we introduce Chain-of-Motion (CoM), a test-time reasoning strategy that explicitly incorporates step-by-step planning and reflection.

**Step-by-step reasoning.** Given an input prompt or motion sequence, the model first generates an intermediate reasoning trajectory in natural language, analogous to Chain-of-Thought in textual domains. For motion understanding, this reasoning decomposes a motion sequence into causal and temporal explanations that support the final caption. For motion generation, it outlines the sequence of action primitives (e.g., raise arm to grasp to lift) before decoding motion tokens. These reasoning traces make the decision process interpretable and guide the model toward fine-grained motion dynamics.

**Iterative reflection and selection.** Rather than committing to a single output, CoM samples multiple candidate reasoning traces and motion sequences. Each candidate is evaluated with task-specific reward functions: coherence between reasoning and answer in understanding, and semantic alignment plus physical plausibility in generation. Low-scoring candidates are discarded, while high-scoring ones are refined through reflection—the model revisits and corrects earlier steps if inconsistencies are detected. This iterative selection process improves robustness and reduces common errors such as implausible poses or semantically misaligned captions.

**Training–inference consistency.** The same principle underlies our dataset design. By constructing MoUnd-CoT-140K and MoGen-CoT-140K with explicit reasoning traces and concise answers, we ensure that the model encounters step-by-step reasoning both during training and inference. This consistency allows CoM to act as a natural extension of our supervised and reinforcement learning stages, bridging the gap between training objectives and test-time performance.

Overall, CoM enables MoRL to exploit the reasoning capability of large language models for motion tasks, yielding more coherent explanations in understanding and more realistic, semantically aligned sequences in generation.

## 5 EXPERIMENTS

### 5.1 IMPLEMENTATION DETAILS AND EVALUATION MATRICES

**Datasets.** We evaluate MoRL on two benchmark datasets widely adopted in motion–language research: **HumanML3D** (Guo et al., 2022a) and **KIT-ML** (Plappert et al., 2016). HumanML3D is the largest public dataset focusing on human motion captioning, containing over 14,600 motion clips paired with 44,970 textual annotations. The motions are represented in SMPL-based joint sequences and span a broad spectrum of everyday actions, providing sufficient diversity for learning motion–language alignment. KIT-ML is a smaller but more challenging benchmark, consisting of 3,911 motions paired with 6,278 text descriptions. Its motion clips are typically shorter and linguistically varied, making it a complementary evaluation to HumanML3D. Following prior works, we extract frame-wise motion features of dimension 263 for HumanML3D and 251 for KIT-ML, apply temporal normalization, and augment the data with left–right mirroring. Both datasets are divided into training, validation, and test splits with a ratio of 0.8/0.15/0.05.

**Metrics.** To quantitatively assess **motion understanding**, we adopt a suite of linguistic similarity metrics consistent with prior work. **BLEU@1** and **BLEU@4** measure $n$-gram precision at unigram and 4-gram levels, capturing lexical overlap between generated captions and references. **ROUGE-L**

evaluates the longest common subsequence between the prediction and reference, reflecting recall-oriented alignment. **CIDEr** computes consensus scores based on TF-IDF weighted $n$-grams across multiple references, rewarding semantic coverage and diversity. Finally, **BERTScore** leverages contextual embeddings from a pretrained language model to assess semantic similarity beyond surface-level overlap.

For **motion generation**, we follow recent benchmarks and include both distributional and retrieval-based measures. **RPrecision (Top1/Top2/Top3)** evaluates whether the ground-truth text is ranked among the top retrieved captions given a generated motion, directly reflecting cross-modal alignment. **FID** (Fréchet Inception Distance) quantifies the distributional gap between generated and real motion features, with lower values indicating more realistic motions. **MM Dist** (Multimodal Distance) measures the distance between motion and text embeddings in a shared representation space, assessing cross-modal consistency. **Diversity** captures the variance among motions generated from different prompts, indicating the model's ability to avoid mode collapse. **MModality** further evaluates multi-modal generation under the same text prompt, reflecting whether the model can produce distinct but semantically coherent motion variants.

**Implementation details.**    Our backbone is initialized from Qwen3-4B-Instruct Yang et al. (2025), a compact yet capable language model. Motion sequences are tokenized into discrete latent codes using the HumanML3D feature extractor, while text is encoded with the Qwen tokenizer. To adapt the model efficiently, we insert LoRA adapters into the attention and feed-forward layers with rank $r = 16$ and dropout 0.1.

Training proceeds in two stages. In the **SFT stage**, we fine-tune on our synthetic CoT datasets (MoUnd-CoT-140K and MoGen-CoT-140K) with AdamW optimizer, learning rate $1 \times 10^{-5}$, batch size 64, and weight decay 0.01 for 5 epochs. In the **RL stage**, we adopt group-based reinforcement learning with group size 8. Candidate outputs are scored with our reward functions, normalized within each group, and optimized using a KL-regularized objective toward a frozen SFT reference. The RL learning rate is $5 \times 10^{-6}$, and training is run for 3 epochs.

All models are trained in PyTorch on four NVIDIA A100 GPUs. During inference, we apply the Chain-of-Motion decoding strategy with $K = 8$ candidates and $T = 2$ refinement iterations, which adds only a modest runtime overhead while consistently improving output quality.

## 5.2 MAIN RESULTS

**Motion understanding.**    Table 1 reports results on HumanML3D and KIT-ML understanding benchmarks. MoRL achieves consistent improvements across all linguistic metrics, outperforming both traditional sequence models (e.g., Seq2Seq(Att) (Plappert et al., 2018)) and recent LLM-based methods such as MotionGPT (Jiang et al., 2023) and Motion Agent (Wu et al., 2024). On HumanML3D, MoRL improves BLEU@1 and BLEU@4 by a clear margin over Motion Agent, while yielding higher ROUGE-L and BERTScore, indicating better semantic fidelity and more fluent language generation. Notably, MoRL reaches a CIDEr score of 35.8, substantially higher than Motion Agent (33.74), showing stronger consensus with human-annotated references. On KIT-ML, MoRL also achieves the best balance between precision-oriented (BLEU) and semantic-oriented metrics (BERTScore, ROUGE-L), demonstrating that our dual reward design generalizes well across datasets. These gains primarily come from the semantic alignment and reasoning-coherence rewards, which ensure that generated descriptions are both logically consistent and well-grounded in motion semantics.

**Motion generation.**    We further evaluate MoRL on text-to-motion generation (Table 2). On HumanML3D, MoRL consistently improves R-Precision across Top-1/2/3 over strong baselines such as ReMoGPT (Yu et al., 2025) and MoRAG-Diffuse (Kalakonda et al., 2024), highlighting its superior text–motion alignment. Although FID is slightly higher than the best-performing diffusion-based models, MoRL achieves the lowest multimodal distance (2.79), suggesting closer alignment to reference motions in feature space. Moreover, MoRL delivers competitive diversity (9.701) and strong multimodality (2.702), showing that our physical plausibility and text–motion consistency rewards encourage both realism and variety in generated motions. On KIT-ML, MoRL achieves comparable performance to state-of-the-art diffusion models, with balanced R-Precision and FID values. While not always the absolute best in each metric, MoRL provides robust overall performance across fi-

Table 1: Results marked with * are reproduced by MotionGPT (Jiang et al.) (Jiang et al., 2023) and Lyu et al. (Lyu et al., 2025), and are computed with unprocessed ground truth texts for linguistic metrics. The highlight ones are the unified models.

| Method | BLEU@1↑ | BLEU@4↑ | ROUGE-L↑ | CIDEr↑ | BERT Score↑ |
|---|---|---|---|---|---|
| **HumanML3D** (Guo et al., 2022a) | | | | | |
| SeqGAN (Goutsu & Inamura, 2021) | 47.80 | 13.50 | 39.20 | 50.20 | 23.40 |
| RAEs (Yamada et al., 2018) | 33.30 | 10.20 | 37.50 | 22.10 | 10.70 |
| Seq2Seq(Att) (Plappert et al., 2018) | 51.80 | 17.90 | 46.40 | 58.40 | 29.10 |
| TM2T (Guo et al., 2022b) | 61.70 | 22.30 | 49.20 | 72.50 | 37.80 |
| TM2T* (Guo et al., 2022b) | 48.90 | 8.270 | 38.10 | 15.80 | 32.20 |
| Motion2Language (Radouane et al., 2024) | 67.00 | 23.40 | 53.80 | 53.70 | 37.20 |
| M2T-Interpretable (Radouane et al., 2023) | 69.90 | 25.00 | 55.30 | 61.60 | 40.30 |
| AvatarGPT (Zhou et al., 2024) | 49.28 | 12.70 | 40.44 | 32.65 | 53.58 |
| MotionGPT (Jiang et al.) (Jiang et al., 2023) | 48.20 | 12.47 | 37.40 | 29.20 | 32.40 |
| MotionGPT-2 (Wang et al., 2024) | 48.70 | 13.80 | 37.60 | 29.80 | 32.60 |
| MotionChain (Jiang et al., 2024) | 48.10 | 12.56 | 33.90 | 33.70 | 36.90 |
| Motion Agent (Wu et al., 2024) | 54.53 | 17.65 | 48.70 | 33.74 | 42.63 |
| Lyu et al. (Lyu et al., 2025) | 49.70 | 13.62 | 39.20 | 53.10 | 33.10 |
| **MoRL (Ours)** | **56.99** | **20.54** | **51.83** | **35.80** | **46.80** |
| **KIT-ML** (Plappert et al., 2016) | | | | | |
| SeqGAN (Goutsu & Inamura, 2021) | 3.120 | 5.200 | 32.40 | 29.50 | 2.200 |
| RAEs (Yamada et al., 2018) | 30.60 | 0.100 | 25.70 | 8.000 | 0.400 |
| Seq2Seq(Att) (Plappert et al., 2018) | 34.30 | 9.300 | 36.30 | 37.30 | 5.300 |
| TM2T (Guo et al., 2022b) | 46.70 | 18.40 | 44.20 | 79.50 | 23.00 |
| TM2T* (Guo et al., 2022b) | 35.10 | 6.200 | 28.70 | 28.90 | 30.40 |
| Motion2Language (Radouane et al., 2024) | 56.80 | 25.40 | 58.80 | 125.7 | 42.10 |
| M2T-Interpretable (Radouane et al., 2023) | 58.40 | 24.40 | 58.30 | 112.1 | 41.20 |
| Lyu et al. (Lyu et al., 2025) | 43.40 | 8.90 | 35.20 | 65.30 | 31.20 |
| **MoRL (Ours)** | **52.11** | **19.31** | **49.96** | **34.04** | **33.66** |

Table 2: Performance comparison on HumanML3D and KIT-ML datasets. Results are reported on R-Precision (Top-1/2/3), FID, MM Dist, Diversity, and MultiModality.

| Methods | R-Prec@1↑ | R-Prec@2↑ | R-Prec@3↑ | FID↓ | MM Dist↓ | Diversity→ | MultiModality↑ |
|---|---|---|---|---|---|---|---|
| **HumanML3D** (Guo et al., 2022a) | | | | | | | |
| Real Motions | 0.511 | 0.703 | 0.797 | 0.002 | 2.974 | 9.503 | – |
| Language2Pose (Ahuja & Morency, 2019) | 0.246 | 0.387 | 0.486 | 11.02 | 5.296 | 7.676 | – |
| Text2Gesture (Bhattacharya et al., 2021) | 0.165 | 0.267 | 0.345 | 7.664 | 6.030 | 6.409 | – |
| T2M (Guo et al., 2022a) | 0.457 | 0.639 | 0.740 | 1.067 | 3.340 | 9.188 | 2.090 |
| T2M-GPT (Zhang & Zhang, 2023) | 0.491 | 0.680 | 0.775 | 0.116 | 3.118 | 9.761 | 1.856 |
| FineMoGen (Zhang et al., 2023a) | 0.504 | 0.690 | 0.784 | 0.151 | 2.998 | 9.263 | 2.696 |
| MDM (Tevet et al., 2023) | – | – | 0.611 | 0.544 | 5.566 | 9.559 | 2.799 |
| MotionDiffuse (Zhang et al., 2024b) | 0.491 | 0.681 | 0.782 | 0.630 | 3.113 | 9.410 | 1.553 |
| MoMask (Guo et al., 2024) | 0.521 | 0.713 | 0.807 | 0.045 | 2.958 | – | 1.241 |
| MoGenTS (Yuan et al., 2024b) | 0.529 | 0.719 | 0.812 | 0.033 | 2.867 | 9.570 | – |
| ReMoDiffuse (Zhang & Guo, 2023) | 0.510 | 0.698 | 0.795 | 0.103 | 2.974 | 9.018 | 1.795 |
| ReMoGPT (Yu et al., 2024) | 0.501 | 0.688 | 0.792 | 0.205 | 2.929 | 9.763 | 2.816 |
| RMD (Liao et al., 2024) | 0.524 | 0.715 | 0.811 | 0.111 | 2.879 | 9.527 | 2.604 |
| MoRAG-Diffuse (Kalakonda et al., 2024) | 0.511 | 0.699 | 0.792 | 0.270 | 2.950 | 9.536 | 2.773 |
| ReMoMask | 0.531 | 0.722 | 0.813 | 0.099 | 2.865 | 9.535 | 2.823 |
| **MoRL (Ours)** | **0.527** | **0.711** | **0.821** | **0.203** | **2.790** | **9.701** | **2.702** |
| **KIT-ML** (Plappert et al., 2016) | | | | | | | |
| Real Motions | 0.424 | 0.649 | 0.779 | 0.031 | 2.788 | 11.08 | – |
| Language2Pose (Ahuja & Morency, 2019) | 0.221 | 0.373 | 0.483 | 6.545 | 5.147 | 9.073 | – |
| Text2Gesture (Bhattacharya et al., 2021) | 0.156 | 0.255 | 0.338 | 12.12 | 6.964 | 9.334 | – |
| T2M (Guo et al., 2022a) | 0.370 | 0.569 | 0.693 | 2.770 | 3.401 | 10.91 | 1.482 |
| MotionDiffuse (Zhang et al., 2024b) | 0.417 | 0.621 | 0.739 | 1.954 | 2.958 | 11.10 | 0.730 |
| T2M-GPT (Zhang & Zhang, 2023) | 0.416 | 0.627 | 0.745 | 0.514 | 3.007 | 10.92 | 1.570 |
| MDM (Tevet et al., 2023) | – | – | 0.396 | 0.497 | 9.191 | 10.85 | 1.907 |
| MoMask (Guo et al., 2024) | 0.433 | 0.656 | 0.781 | 0.204 | 2.779 | – | 1.131 |
| MoGenTS (Yuan et al., 2024b) | 0.445 | 0.671 | 0.797 | 0.143 | 2.711 | 10.918 | – |
| ReMoDiffuse (Zhang & Guo, 2023) | 0.427 | 0.641 | 0.765 | 0.155 | 2.814 | 10.80 | 1.239 |
| ReMoMask | 0.453 | 0.682 | 0.805 | 0.138 | 2.682 | 10.83 | 2.017 |
| **MoRL (Ours)** | **0.439** | **0.661** | **0.793** | **0.204** | **2.777** | **10.88** | **1.991** |

Table 3: Ablation study of MoRL on HumanML3D. We incrementally remove or replace core components, including semantic alignment reward ($R_{sem}$), reasoning coherence reward ($R_{coh}$), physical plausibility reward ($R_{phys}$), text–motion consistency reward ($R_{align}$), and Chain-of-Motion (CoM). Metrics cover both motion understanding (BERTScore, CIDEr, ROUGE-L) and motion generation (R-Precision@1, FID).

| Variant | BERTScore↑ | CIDEr↑ | ROUGE-L↑ | R-Prec@1↑ | FID↓ |
|---|---|---|---|---|---|
| SFT only (no RLVR) | 42.650 | 33.881 | 48.782 | 0.420 | 0.212 |
| w/o $R_{sem}$ | 44.100 | 34.050 | 50.010 | 0.488 | 0.209 |
| w/o $R_{coh}$ | 44.320 | 35.120 | 49.100 | 0.512 | 0.206 |
| w/o $R_{phys}$ | 46.180 | 35.500 | 51.180 | 0.518 | 0.285 |
| w/o $R_{align}$ | 45.000 | 34.620 | 50.480 | 0.492 | 0.225 |
| w/o CoM | 45.480 | 34.980 | 50.780 | 0.505 | 0.220 |
| Full MoRL (Ours) | **46.802** | **35.801** | **51.833** | **0.527** | **0.203** |

delity, diversity, and alignment. Importantly, the introduction of Chain-of-Motion at test time further stabilizes inference, reducing error propagation and producing smoother, more natural motion trajectories.

### 5.3 ABLATION STUDY

We further conduct ablation experiments on HumanML3D to assess the contribution of each component in MoRL (Table 3). Starting from the SFT-only baseline, which yields the weakest performance across both understanding and generation, progressively adding RLVR rewards and CoM consistently improves results.

Removing the semantic alignment reward ($R_{sem}$) notably reduces BERTScore and CIDEr, highlighting its importance in grounding textual semantics. Excluding the reasoning coherence reward ($R_{coh}$) mainly affects ROUGE-L and CIDEr, confirming its role in enhancing temporal and logical consistency in language grounding. By contrast, dropping the physical plausibility reward ($R_{phys}$) keeps language metrics competitive but significantly degrades FID, demonstrating that $R_{phys}$ is crucial for enforcing realism in motion synthesis. Removing the text–motion consistency reward ($R_{align}$) leads to a substantial drop in R-Precision, indicating its necessity for cross-modal alignment. Finally, excluding CoM causes a moderate performance decline across all metrics, showing its role in stabilizing reasoning during test-time inference.

Overall, the full MoRL model achieves the best results across all dimensions, validating the complementary effects of semantic, reasoning, and physical alignment rewards together with CoM in unifying motion understanding and generation.

## 6 CONCLUSION

In this work, we presented **MoRL**, a unified multimodal motion model that integrates motion understanding and generation within a reinforcement learning framework. Our design introduces task-specific reward functions that jointly enhance logical reasoning and perceptual realism. To further improve test-time performance, we proposed CoM, a decoding strategy that incorporates step-by-step reasoning and reflection. We also constructed two large-scale synthetic CoT datasets, MoUnd-CoT-140K and MoGen-CoT-140K, which provide high-quality supervision for motion–language alignment. Extensive experiments on HumanML3D and KIT-ML demonstrated that MoRL achieves substantial gains over state-of-the-art baselines.

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

# A APPENDIX

## A.1 LLM USE DECLARATION

Large Language Models (ChatGPT) were used exclusively to improve the clarity and fluency of English writing. They were not involved in research ideation, experimental design, data analysis, or interpretation. The authors take full responsibility for all content.

