# OpenReview forum: "MoRL: Reinforced Reasoning for Unified Motion Understanding and Generation"
_ICLR.cc/2026/Conference — ICLR 2026 Conference Withdrawn Submission_

### Official Review · Reviewer_ErN1 · 2025-10-31

**Soundness:** 2
**Presentation:** 2
**Contribution:** 2
**Rating:** 2
**Confidence:** 4

**Summary:**

This paper presents MoRL, a unified framework for human motion understanding and generation, leveraging a multi-modal large language model enhanced by reinforcement learning and step-by-step reasoning strategies. The approach introduces specific rewards for semantic alignment, reasoning coherence, physical plausibility, and text-motion consistency, aiming to improve both the interpretability and realism of generated motions. MoRL is trained and evaluated on large human motion-language datasets, and is complemented by a new Chain-of-Motion (CoM) test-time inference scheme and two synthetic chain-of-thought (CoT) motion datasets (MoUnd-CoT-140K and MoGen-CoT-140K). Experiments on benchmark datasets demonstrate gains over a wide range of baselines, and ablation studies dissect the value of individual model components.

**Strengths:**

- MoRL presents an integrated approach for both motion understanding and generation, offering a single framework that closes the gap between perception and synthesis in motion-language modeling.
- The reward functions go beyond prior generic similarity metrics: the logical coherence reward for reasoning traces and explicit physical plausibility for motion generation are clear step-ups over most existing motion-language systems.
- Quantitative results across linguistic (BLEU, ROUGE, CIDEr, BERTScore) and distributional metrics (R-Precision, FID, MM Dist, Diversity) demonstrate consistent improvements.

**Weaknesses:**

- **Lack of supplementary materials and qualitative evidence (visualizations/videos)**： The submission provides neither a supplementary document nor qualitative visualizations (e.g., trajectory plots, attention/activation maps, sample reasoning traces aligned with frames) or videos of generated motions. For a generative motion system, the absence of side-by-side videos against baselines and ablations makes it difficult to assess realism, physical plausibility, temporal coherence, and failure modes, which **materially undermines credibility**. Please include a supplementary PDF and a project page with videos (paired with CoM/CoT reasoning snippets), as well as qualitative comparisons to Motion-R1/MotionRL.
- **Novelty Relative to Recent Work**: While the reinforcement learning and CoT/CoM techniques are assembled well, the combination does not significantly depart from recent trends exemplified by MotionRL, Motion-R1, and Open-Reasoner-Zero; the primary novelty appears incremental rather than foundational. This modest advancement is particularly notable considering the reliance on recently published reinforcement learning pipelines with similar reward shaping and reflection-style reasoning.
- **Reward Implementation Ambiguities**: The reinforcement learning reward structure is stated clearly but lacks necessary specifics. For example, details on normalization strategies, the precise form of the entailment model $f_{\mathrm{NLI}}$, the codebook size ($N$) and its impact on representation granularity, and hyperparameter $\lambda_1$, $\lambda_2$ selections for physical plausibility rewards are omitted (see Section 4.4, page 5-6). This hampers reproducibility and undermines claims regarding robustness and optimality.
- **Inadequate Differentiation from Motion-R1 and MotionRL**: Both these works (see Related Work, page 2) have explored RL-based and CoT-infused motion generation. It's unclear to what extent MoRL substantially improves over or is distinct from these baselines, especially lacking direct, head-to-head empirical or qualitative comparison in Table 1 or Table 2 (no explicit Motion-R1 or MotionRL entries).

**Questions:**

1. Can the authors clarify the selection and training of the NLI model $f_{\mathrm{NLI}}$ used for the reasoning coherence reward? Is it frozen or fine-tuned? What is its domain (textual or motion-natural language mixed)?
2. Are there examples or a qualitative analysis of the generated reasoning traces for typical and failure cases? How do they compare in logical structure and relevance to those produced by recent LLM or multimodal CoT-based baselines?
3. Could the authors provide more details about the quantization mechanism in the motion tokenizer, including how ambiguous cases or codebook collisions are handled?

---

### Official Review · Reviewer_Dwdq · 2025-11-01

**Soundness:** 3
**Presentation:** 2
**Contribution:** 3
**Rating:** 4
**Confidence:** 2

**Summary:**

The paper presents a unified multimodal motion model that handles both motion understanding and text-to-motion generation, it combines supervised CoT tuning with reinforcement learning. The reward design is dual-headed: semantic alignment + reasoning coherence for understanding, and physical plausibility + text–motion consistency for generation. MoRL uses a Qwen3-4B backbone with a VQ-VAE motion tokenizer, and is optimized via a GRPO-style objective with KL regularization. On HumanML3D and KIT-ML, MoRL reports consistent gains compared to extensive baselines, with ablations showing each reward and CoM contribute additively.

**Strengths:**

- The paper unifies motion understanding and generation with task-specific rewards. The paper adopts a simple but effective RL recipe, using GRPO-style group sampling with KL to a frozen reference avoids heavy heuristics yet yields consistent gains.

- The work has strong CoT data engine. Two large CoT datasets align motions with reasoning traces and concise answers, providing good supervision for both directions.

- Evaluations cover comprehensive metrics for both understanding and generation comparing with extensive baselines.

**Weaknesses:**

- Results are only on HumanML3D and KIT-ML, harder and more diverse settings (like long sequences and multi-person) aren’t covered.

- CoM samples K=8 candidates with T=2 refinement, the paper calls overhead “modest” but reports no latency/throughput numbers.

- Headline gains of experimental results are 4.17% BERT and 3% FID, and MoRL is not best on all metrics (like FID vs diffusion baselines).

**Questions:**

- Could the author provide a human study for realism and text–motion alignment to complement automatic metrics?

- Could the authors report end-to-end inference latency/throughput with CoM vs single-pass decoding?

- The VQ-VAE motion tokenizer lacks concrete hyperparameters like codebook size that affect generation quality.

---

### Official Review · Reviewer_VD7i · 2025-11-01

**Soundness:** 3
**Presentation:** 4
**Contribution:** 3
**Rating:** 8
**Confidence:** 4

**Summary:**

This paper proposes MoRL, a unified model for human motion understanding and generation trained with SFT and RL. The approach has three main components: (1) task-specific reward functions combining semantic alignment and reasoning coherence for understanding (2) two synthetic Chain-of-Thought datasets (MoUnd-CoT-140K and MoGen-CoT-140K) created using Gemini to align motion sequences with reasoning traces (3) Chain-of-Motion (CoM), a test-time decoding strategy with iterative refinement. Experiments on HumanML3D and KIT-ML show improvements over baselines; ablations isolate contribution of each reward and of CoM.

**Strengths:**

- well motivated problem. Motion understanding and generation have been studied separately but unifying them with a shared representation is valuable. Table 3 shows that joint training improves both understanding and generation, suggesting these tasks reinforce each other.
- the four reward (semantic, coherence, physical, alignment) all cover complementary aspects and I appreciate the ablations in table 3 to show each contributes meaningfully
- strong experimental validation with two benchmarks with comprehensive metrics + comparison with over 15 baselines
- another strength is the large-scale dataset contributions with both MoUnd-CoT-140K and MoGen-CoT-140K providing valuable resources for future research
- CoM is a very interesting idea and I'm glad to see it worked out. Its success shows that we can adapt the paradigm of test-time compute scaling from LLMs to motion domain, showing its generalizability

**Weaknesses:**

-Missing an analysis of the quality of the synthetic CoT data as there is no human evaluation of the dataset provided. We have no idea which gemini reasoning traces are actually correct. I hope the authors can share more data quality metrics and analysis.
- I would like to see when does MoRL fail. what types of motions or captions are challenging. A qualitative and quantitative error analysis would be nice.

**Questions:**

- Why did you use GRPO specifically? Can you compare with PPO, DPO, and SFT on CoT data w/o RL?
- a 4B model size seems small. Did you try smaller and larger models and see how it affects performance?
- The paper states that SFT uses MoUnd-CoT-140K and MoGen-CoT-140K, but doesn't explicitly specify which dataset is used for the RL stage. I would imagine MoUnd-CoT-140K and MoGen-CoT-140K would also be used for RL, but then Section 5.1 mentions "both datasets are divided into training, validation, and test splits with a ratio of 0.8/0.15/0.05," referring to HumanML3D and KIT-ML, which suggests these might be used for training. Is this correct? Can you in general clarify which datasets are used at what point in the training pipeline?

---

### Official Review · Reviewer_oK66 · 2025-11-02

**Soundness:** 2
**Presentation:** 3
**Contribution:** 2
**Rating:** 4
**Confidence:** 3

**Summary:**

This paper introduces MoRL, a unified multimodal motion model designed to enhance human motion understanding and generation through both supervised fine-tuning and reinforcement learning with verifiable, task-specific rewards. The reward framework integrates semantic and reasoning coherence for understanding tasks and physical plausibility with text–motion consistency for generation tasks, thereby improving logical reasoning and perceptual realism. To strengthen test-time reasoning, the authors propose Chain-of-Motion (CoM), a method that supports step-by-step planning and reflection during inference. They also build two large-scale reasoning datasets—MoUnd-CoT-140K and MoGen-CoT-140K—linking motion sequences to reasoning traces and action descriptions. Experiments on HumanML3D and KIT-ML demonstrate that MoRL outperforms state-of-the-art baselines in both understanding and generation tasks.

**Strengths:**

1. The paper presents a unified multimodal framework that effectively integrates semantic, reasoning, and physical consistency rewards, leading to more logically coherent and perceptually realistic motion understanding and generation.
2. It introduces Chain of Motion reasoning and large-scale CoT datasets, which significantly enhance the model’s interpretability and performance, achieving measurable improvements over state-of-the-art baselines on standard benchmarks.
3. The paper is well-written, presenting a complex technical system with conceptual clarity and a logical narrative that is easy to follow.

**Weaknesses:**

1. The proposed method fails to outperform or at least reach comparable performance to all the baselines, especially in terms of CIDEr, where it was significantly surpassed by certain baselines.
2. In the ablation study, it seems that the performance of full MoRL and that without certain reward item or CoM are comparable. Moreover, it does not even show a significant improvement of full MoRL over SFT only.

**Questions:**

I am curious about the approximate probability of triggering the iterative reflection and selection process in CoM. Typically, how many rounds does it take to successfully generate a result? During the reflection process, does the model produce new modalities of behavior, or does it mainly modify inconsistent details?

---

### Note · Authors · 2025-11-29

I have read and agree with the venue's withdrawal policy on behalf of myself and my co-authors.